# Sources of Gain:
# Decomposing Performance in Conditional Average Dose Response Estimation

## Abstract

Estimating conditional average dose responses (CADR) is an important but challenging problem. Estimators must correctly model the potentially complex relationships between covariates, interventions, doses, and outcomes. In recent years, the machine learning community has shown great interest in developing tailored CADR estimators that target specific challenges. Their performance is typically evaluated against other methods on (semi-) synthetic benchmark datasets. Our paper analyses this practice and shows that using popular benchmark datasets without further analysis is insufficient to judge model performance. Established benchmarks entail multiple challenges, whose impacts must be disentangled. Therefore, we propose a novel decomposition scheme that allows the evaluation of the impact of five distinct components contributing to CADR estimator performance. We apply this scheme to eight popular CADR estimators on four widely-used benchmark datasets, running nearly 1,500 individual experiments. Our results reveal that most established benchmarks are challenging for reasons different from their creators' claims. Notably, we find that confounding - the key challenge that motivated recent methods - does not significantly affect CADR estimation performance for the considered datasets. We discuss the major implications of our findings and present directions for future research.

## 1 Introduction

Despite the surge in machine learning (ML) methods for estimating heterogeneous treatment effects (Shalit et al., 2016; Johansson et al., 2016; Louizos et al., 2017; Shi et al., 2019; Yoon et al., 2018; Johansson et al., 2020; Wager & Athey, 2018; Hill, 2011), there is comparatively little research on estimating the heterogeneity of dose responses, i.e., the responses to interventions with a continuous component. This is surprising as such interventions are ubiquitous and understanding a unit's response to them is critical in several domains, e.g., for assigning optimal discounts in marketing (Miller & Hosanagar, 2020), or for administering an effective dose of a medication (Frei & Canellos, 1980). Estimating dose responses from observational data is distinct from estimating treatment effects: units can be exposed to one of several different interventions for which the associated dose can vary across units. This introduces several unique challenges and calls for tailored methodologies.

The literature proposing ML-estimators for conditional average dose responses (CADR), also referred to as "individual" (Schwab et al., 2019) or "heterogeneous" (Zhu et al., 2024) dose responses, is confined to only a few methods which were proposed in the past five years (Schwab et al., 2019; Bica et al., 2020; Nie et al., 2021; Wang et al., 2022; Zhang et al., 2022; Zhu et al., 2024; Nagalapatti et al., 2024; Kazemi & Ester, 2024), and that have not yet seen wide-spread usage in real-world applications. While the state of the art has progressed significantly, research lacks alignment, focusing on different challenges and using different benchmarking datasets. This becomes especially apparent when reviewing the established benchmarking practices in CADR estimation: to date, the field has relied on a selection of (semi-) synthetic benchmarking datasets created from manually defined data-generating processes (DGPs). These datasets claim to test estimators in the presence of cer-

tain challenges, most notably "confounding".[1] Prior work states confounding as the key challenge motivating their method, but they do not clarify how exactly confounding makes CADR estimation challenging. Conversely, as our experiments show, those DGPs expose estimators to more than just one challenge, of which confounding is not the most important one.

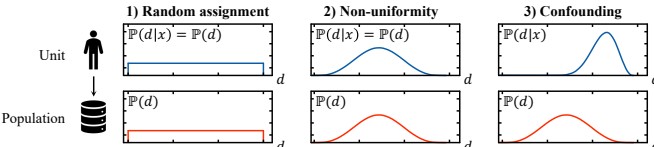

Figure 1: **Selected components of our decomposition scheme.** To disentangle the effects of confounding from the effects of non-uniform distributions of doses, we evaluate estimators in three scenarios: 1) When doses are randomly sampled from a uniform distribution, 2) when those distributions are not uniform, but also not specific to a certain unit, and 3) when the data is confounded, so when dose assignment is specific to a certain unit. The distribution of doses across the total population is the same in steps 2) and 3). Our complete scheme includes two additional steps related to the distribution of interventions when there are multiple intervention options (cf. Section 4).

We believe that further progress in the field requires a deeper understanding of the nature of CADR estimation and the challenges therein. To that end, we propose a problem formulation for CADR estimation that unifies existing research. We conceptualize DGPs along this formulation and identify five components contributing to CADR estimator performance. To facilitate future research, we propose a novel decomposition scheme that disentangles performance along these five components, allowing researchers to understand the sources of model performance (cf. Figure 1).

To that end, our paper makes three important contributions: (1) We introduce a unifying problem formulation for CADR estimation and conceptualize synthetic DGPs for benchmarking dataset creation; (2) We propose a scheme for decomposing model performance along the mechanisms of a DGP and specific challenges in CADR estimation; (3) We test a selection of ML estimators and decompose their performance on four of the most popular benchmark datasets for CADR estimation. We elicit strengths and weaknesses and compare their performance against traditional supervised learning algorithms. As such, we aim to establish a standardized approach that facilitates an effective evaluation and comparison of methods. Our ambition is for the proposed decomposition scheme to be adopted in future work on CADR estimation and to guide future research.

**Outline.** Section 2 introduces related work on evaluating ML estimators, and previous practices in decomposing model performance. We conceptualize CADR estimation in Section 3. In Section 4, we conceptualize DGPs and introduce our novel decomposition scheme. We illustrate our approach for a prominent dataset from Bica et al. (2020) in Section 5, introducing established methods and discussing the findings. We provide results on different datasets in Section 6 and conclude in Section 7.

## 2 CONTEXT

**Evaluating conditional average intervention response estimators is challenging.** Evaluating the performance of conditional average intervention response estimators is challenging, as per unit of interest we can only observe the response to a single "factual" intervention, and never to any other one "counterfactual" one (Holland, 1986). In conditional average treatment effect (CATE) estimation, there is only one counterfactual intervention: units have received either the treatment or the control. In CADR estimation, this is further complicated. First, one can apply one of several distinct interventions, and second, a practically infinite number of doses. The full dose response curve hence cannot be observed. In consequence, using observational data limits evaluation to measuring accuracy in predicting factual responses as in, e.g., supervised learning (Hastie et al., 2017).

**Lack of established benchmarking practices.** In ML research, CADR estimators are typically evaluated using semi- or fully synthetic datasets that are specified by a certain DGP. This DGP

---

[1]Some ML papers refer to confounding as "selection bias" which also relates to the out-of-sample generalizability of response estimates (Haneuse, 2016). To reduce ambiguity, we adopt the terminology traditionally used in the causal inference literature (Pearl, 2022; Angrist & Pischke, 2009; Imbens & Rubin, 2015; Cunningham, 2021).

allows the calculation of counterfactual responses of any unit and to any intervention, to overcome the challenges in evaluating using observational data. In CATE estimation researchers have relied predominantly on a single dataset for evaluating estimator performance (Curth et al., 2021). For CADR estimation, there is no such established standard, and researchers have relied on several different datasets. Moreover, the creators of these datasets typically do not clarify the challenges present that might complicate CADR estimation. We will show in our experiments that a dataset often embodies several challenges (cf. Section 4), which further hinders the understanding of model performance. We aim to alleviate this issue, by proposing a scheme to decompose performance along different aspects of benchmarking data.

**Decomposing model performance.** It is common practice in ML research to decompose (or to "ablate") complex model architectures by systematically adding and removing components to measure their impact on performance. In CADR estimation, such studies have, e.g., been conducted by Schwab et al. (2019) and Bica et al. (2020). While such ablations inform about which part of an architecture contributes to improvements in model performance, they do not allow us to understand the challenges in a certain dataset. This understanding is critical to effective methodology selection in real-life applications. We attempt to close this gap in CADR estimator research by providing a scheme to decompose datasets, not models. Such a decomposition enables us to understand the scenarios in which an estimator might work well or fail. This data-centric decomposition of model performance is in line with calls in other ML research domains (Ye et al., 2022; Yang et al., 2022).

## 3 PROBLEM FORMULATION

**Defining CADR estimation.** We find varying definitions of CADR estimation in the literature, typically differing in the availability of only one (Hirano & Imbens, 2004; Nie et al., 2021) or several (Schwab et al., 2019; Bica et al., 2020) distinct interventions with an associated dose. In the following, we propose a unifying framework that subsumes any of these definitions.

We leverage the Neyman-Rubin potential outcomes (PO) framework (Rubin, 1974; Splawa-Neyman et al., 1990) and expect a unit of interest $i$ to be specified by a realization $\mathbf{x}_i$ of random variable $\mathbf{X} \in \mathcal{X}$ with $\mathcal{X} \subset \mathbb{R}^m$ being the $m$-dimensional feature space. The unit is exposed to a single intervention $t_i$ sampled from $T \in \mathcal{T}$, with the intervention space $\mathcal{T} = \{\omega_1, \ldots, \omega_k\}$ being discrete with $k$ different intervention options. Every unit is also exposed to a continuous intensity or dose $d_i$ sampled from $D \in \mathcal{D}$ with $\mathcal{D} \subset \mathbb{R}$. For the remainder of our paper and without loss of generality, we set $\mathcal{D} = [0, 1]$. For any realization of intervention and dose variables, there is a potential outcome $Y(t, d) \in \mathcal{Y} \subset \mathbb{R}$. In line with Schwab et al. (2019) we define $Y(t, d)$ as the "dose response".

We are interested in finding an estimate of the conditional average dose response (CADR), defined as

$$\mu(t, d, \mathbf{x}) = \mathbb{E}[Y(t, d)|\mathbf{X} = \mathbf{x}] \tag{1}$$

for every $t \in \mathcal{T}$, $d \in \mathcal{D}$ and $\mathbf{x} \in \mathcal{X}$, which is in line with the definition by Bica et al. (2020). A detailed overview of the notation is presented in Appendix A.

**Understanding the causal structure of dose responses.** The challenges in estimating dose responses arise from the causal relationships between variables $\mathbf{X}$, $D$, $T$, and $Y$, which we illustrate in a single-world intervention graph (SWIG, Richardson & Robins, 2013) in Figure 2. Typically, an intervention $t$ is chosen based on the observed realization of the covariates $\mathbf{x}$. Given the intervention, a dose $d$ is assigned. The observed potential outcome is subsequently dependent on the realization of all those variables $\mathbf{x}$, $t$, and $d$. To find an unbiased estimate of $\mu(\cdot)$ we must hence use an estimator that is flexible enough to capture the relationship between outcome and intervention variables, while correctly adjusting for the effects of $\mathbf{X}$ on all $Y$, $T$, and $D$ (Nie et al., 2021), but also for the effect of $T$ on $D$. The simultaneous influence of $\mathbf{X}$ on the intervention variables and the response is referred to as "confounding" (Porta, 2014).

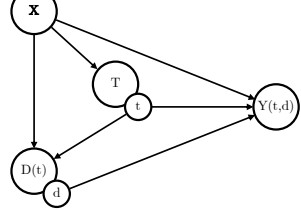

Figure 2: The **SWIG** represents the causal dependencies between variables in observational data for CADR estimation. Covariates $\mathbf{x}$ influence both intervention type $t$ and dose $d$. The dose is also influenced by $t$. Outcome $y$ depends on all $\mathbf{x}$, $t$, and $d$.

**On the identifiability of CADR.** Estimating dose responses from observational data relies on a set of assumptions (Stone, 1993). Specifically, the identifiability of intervention responses requires "strong ignorability" (Rosenbaum & Rubin, 1983), subsuming "consistency", "no hidden confounders", and "overlap". We provide definitions of these assumptions in Appendix C. We assume these assumptions to hold for the remainder of our work, as most previously proposed methods require them. On top of that, different from estimating treatment effects (Imbens, 2004; Petersen et al., 2012), there is little research on the impacts of violations of these assumptions when estimating dose responses.

## 4 A DECOMPOSITION SCHEME FOR PERFORMANCE OF CADR ESTIMATORS

**Abstracting DGPs.** When working with observational data, the underlying natural DGP is typically partially or fully unknown, adding to the challenge of evaluating CADR estimator performance (cf. Section 2). To counter this, ML research typically relies on synthetic DGPs for benchmarking intervention response estimators. Those DGPs assume a causal structure (in our case the SWIG in Figure 2) and define relationships between different variables, allowing for full control over challenges in the data, such as confounding. A typical DGP for CADR consists of four components: (1) the definition of observed units through a covariate vector $\mathbf{X}$, (2) an intervention assignment function $a_t(\cdot)$ assigning an intervention to every unit, (3) a dose assignment function $a_d(\cdot)$ assigning a dose, and (4) an outcome function $\mu(\cdot)$. Each of those components influences the resulting benchmarking dataset. Functions can be defined to take variable inputs. By taking as input the covariate vector $\mathbf{X}$, $a_t(\cdot)$ and $a_d(\cdot)$ introduce confounding in the resulting data. While sharing a unified causal structure, the existing benchmarking datasets are diverse in the mechanisms by which each element of the DGP is defined. This introduces ambiguity regarding the impacts of each element of the DGP on model performance.

We provide a list of the established CADR benchmarking datasets in Appendix F along with an overview of established CADR estimators, and the datasets used for their evaluation. The most widely used datasets are those proposed in Bica et al. (2020) (TCGA-2) and Nie et al. (2021) (IHDP-1, News-3, and Synth-1). Typically, assignment functions are non-deterministic to comply with the strong ignorability assumption. For interventions, this is done by assigning non-zero probabilities to the various possible interventions before sampling a factual intervention per unit. Doses are sampled from a distribution with a strictly positive probability mass over $\mathcal{D}$ and a mode conditional on the covariates of a unit, so, e.g., from a normal distribution (Nie et al., 2021) or a beta distribution (Bica et al., 2020). The outcome function $\mu(\cdot)$ specifies a CADR by taking as input the resulting vectors $\mathbf{X}$, $D$, and $T$. Individual responses are generated by adding random noise.

**Disentangling impacts of synthetic DGPs on estimator performance.** Confounding is the presence of a covariate vector that influences intervention and dose assignment, as well as the response of a unit. Different mechanisms have been proposed to introduce confounding in a benchmarking dataset: Bica et al. (2020) set a modal intervention and dose that would maximize a unit's CADR, whereas Nie et al. (2021) use some polynomial non-linear function mapping a unit's covariates to a modal dose. The level and complexity of confounding in a (semi-) synthetic dataset may vary, depending on the confounding mechanism in the synthetic DGP. However, only some of the established DGPs allow to vary the level of confounding. Therefore, it may be unclear what exactly is driving model performance: the ability of a method to model a complex CADR, as specified by $\mu(\cdot)$, or its ability to adjust for confounding.

Moreover, the assignment functions $a_t(\cdot)$ and $a_d(\cdot)$ introduce more challenges to the dataset than just confounding. As a certain assignment mechanism impacts the probability of being assigned different interventions and doses per unit, it simultaneously influences their distribution across the observed population (cf. Figure 1 and a detailed discussion in Appendix B). This is most evident upon analyzing benchmarking datasets with tunable levels of confounding, such as the TCGA-2 dataset proposed by Bica et al. (2020). When the level of dose confounding is set to $\alpha = 1$, so no confounding, doses are uniformly

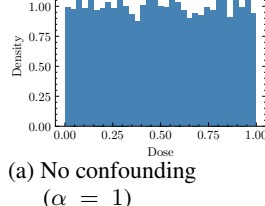

(a) No confounding ($\alpha = 1$)

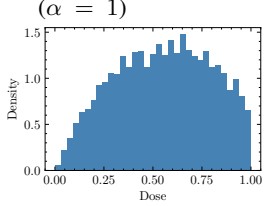

(b) Confounding ($\alpha = 2$)

Figure 3: **Dose distribution** for different levels of confounding in data by Bica et al. (2020)

distributed across $\mathcal{D}$. When the level of confounding increases, the distribution of doses changes (cf. Figure 3). This leads to significantly fewer observations for some dose intervals, potentially impacting the performance of ML estimators (Kokol et al., 2022).

**Decomposing performance by randomizing intervention assignment.** The adjustment for confounding is a key challenge in CADR estimation (Bica et al., 2020; Nie et al., 2021). Yet, without further investigation, performance on a dataset could be attributed to any of the above-mentioned challenges in the data. Standard ML benchmarking practices do not reveal whether the performance of an estimator is impacted by non-linearity of responses, confounding factors, or from the intervention and dose distributions across the population. To overcome this limitation, we propose a novel decomposition scheme for CADR estimator performance. Our scheme is performance metric-agnostic. The choice of performance metric is arbitrary and situation-specific.

We consider two boundary scenarios: In the "randomized" scenario, interventions and doses are completely randomized and sampled uniformly. In the "non-randomized" scenario, the data aligns with its creators' specifications, where interventions and doses adhere to assignment functions $a_t(\cdot)$ and $a_d(\cdot)$. We then explore three intermediary scenarios that progressively move from the "randomized" to the "non-randomized" setup.

First, we generate scenario "$t$ non-uniformity". Per unit, an intervention is sampled from a joint distribution that is equivalent to the distribution of interventions in the "randomized" scenario. This conduct maintains the effects of $a_t(\cdot)$ on the distribution of interventions across the total population, yet removes confounding, as $\mathbb{P}(t|\mathbf{x}) = \mathbb{P}(t)$ for all $t \in \mathcal{T}$ and $\mathbf{x} \in \mathbf{X}$. This scenario informs about the impact of population-level changes in intervention distributions. Second, we generate scenario "$t$ confounding", in which we operate under random dose assignment, but confounded interventions as generated by $a_t$. This allows us to isolate the effects of intervention confounding from distributional effects investigated previously. Third, and in addition to intervention confounding, we repeat the process for dose assignment, generating scenario "$d$ non-uniformity", in which $\mathbb{P}(d|\mathbf{x}) = \mathbb{P}(d)$ for all $d \in \mathcal{D}$ and $\mathbf{x} \in \mathbf{X}$.. The final scenario adds dose confounding, by generating doses according to $a_d$, yielding the "$d$ confounding" or "non-randomized" scenario, which aligns with the original specifications of the data. To be in line with the causal dependencies in observational data as outlined in Section 3, we chose to first decompose along the treatment assignment, yet our scheme is flexible and would allow for alterations. The full decomposition scheme is summarized in Algorithm 1 in Appendix E. We refer to Appendix J for the technical implementation. We summarize the resulting five scenarios below:

1. (*Randomized*) Random interventions and doses (sampled from uniform distributions)

2. (*t non-uniformity*) Non-uniformly distributed interventions and random doses

3. (*t confounding*) Confounded interventions and random doses

4. (*d non-uniformity*) Confounded interventions and non-uniformly distributed doses

5. (*Non-randomized / d confounding*) Confounded interventions and confounded doses

By iteratively changing key characteristics of the data, our decomposition scheme represents an experimental design to test for the effects of various contributing factors on CADR estimator performance. As such, we attempt to further bridge between benchmarking practices in ML and experimental study design in the wider field of causal inference (Rubin, 2008; Shadish et al., 2015).

## 5 CASE STUDY: DECOMPOSING PERFORMANCE ON THE TCGA-2 DATASET

### 5.1 EXPERIMENTAL SETUP

**Dataset.** To demonstrate the workings of our decomposition scheme and how it helps to understand model performance, we apply it to the TCGA-2 dataset proposed by Bica et al. (2020).[2] Several other benchmarking datasets for CADR estimation have been proposed in previous studies, which we enlist in Appendix F. We selected the TCGA-2 dataset for several reasons. First, it comprises the assignment of one of three distinct interventions per unit and an associated dose, whereas most other datasets consider a single intervention. Second, the covariate matrix used in the TCGA data (Cancer Genome Atlas Research Network et al., 2013) is frequently used in other research. Third, the assignment mechanisms used in the DGP find use in several succeeding papers, such as in Nie et al. (2021), Schweisthal et al. (2023), Nie et al. (2021), and Vanderschueren et al. (2023). For technical details, we refer to the original paper (Bica et al., 2020).

**ML methods for CADR estimation.** We decompose the performance of selected established CADR estimators, as well as several supervised estimators, to provide a broad set of baseline methods. Hereby, we follow calls from related ML research to test novel algorithms against a wider array of methods (Qin et al., 2020). Much of recent ML research has been devoted to developing neural network architectures to tackle analytical problems, with many papers ignoring classical methods such as regression or tree-based approaches. First, we apply three prominent ML estimators for CADR estimation, namely DRNet (Schwab et al., 2019), SCIGAN (Bica et al., 2020), and VCNet (Nie et al., 2021). Each of these is a neural network (Goodfellow et al., 2016), providing several favorable characteristics to modeling data (Hornik et al., 1989). The estimators have been the first tailored ML methods for dose response estimation, and have been used as benchmark methodologies for several later-proposed methods. Each method uniquely tackles CADR estimation: SCIGAN uses generative adversarial networks (GANs, Goodfellow et al., 2020) to generate additional counterfactual outcomes per observed unit and effectively randomize intervention and dose assignment, as such removing confounding. DRNet trains separate models on a shared representation learner per combination of dose intervals and interventions, to reinforce the influence of the intervention variables in the model. VCNet follows a similar motivation, yet leverages a varying-coefficient architecture (Hastie & Tibshirani, 1993) to accomplish this. Appendix D provides a detailed description of the three methods. A complete overview of ML dose response estimators is provided in Appendix F. Following calls from other fields in ML to benchmark novel methodologies against a complete set of established methods (Qin et al., 2020), we further apply five supervised learning methods, which have not been sufficiently benchmarked in prior work (for a list of benchmark methodologies per established ML dose response estimator see again Appendix F). We apply a linear regression model, a regression tree (Breiman et al., 2017), a generalized additive model (GAM, Wood, 2017), xgboost (Chen & Guestrin, 2016) as a state-of-the-art implementation of a gradient-boosted decision tree (Friedman, 2001), and a simple feed-forward multilayer perception (MLP). In comparing ML CADR estimators with traditional supervised learning methods we aim to understand both the complexity of benchmarking datasets and differences in the performance of methods concerning the challenges in dose response estimation. This facilitates practitioners and researchers in making a conscious tradeoff between interpretability and performance of estimators (Bell et al., 2022), as some of the targeted estimators introduce significant complexity over established techniques.

**Performance evaluation.** Our decomposition scheme is agnostic to the selection of a performance metric and could be used across scenarios and use cases. Next to CADR estimators, it could similarly be used to evaluate "average" dose response estimators. For evaluating CADR estimator performance, Schwab et al. (2019) propose three performance metrics, especially the "policy error" and "dose policy error" which evaluate the capability of an estimator to identify the most effective interventions and doses by the magnitude of the response, as well as the "mean integrated squared error" (MISE), which evaluates the mean accuracy of the CADR estimate over the intervention and dose spaces. Since the MISE makes minimal assumptions about the domain of application or use of a CADR estimator, we adopt it in the experiments reported in this paper. For a number of test units

---

[2]The level of intervention and dose confounding can be varied in the TCGA-2 dataset. We opt for the standard values proposed in the original paper (Bica et al., 2020) with the level of intervention confounding set to $\kappa = 2$ and the level of dose confounding set to $\alpha = 2$.

Table 1: **Performance decomposition on TCGA-2 dataset (Bica et al., 2020)**

| Method | Scenario | | | | |
|---|---|---|---|---|---|
| | random. | $\rightarrow t$ non-unif. $\rightarrow$ | $t$ conf. | $\rightarrow d$ non-unif. $\rightarrow$ | $d$ conf. |
| Lin. reg. | 4.74 ± 0.02 | 4.84 ± 0.03 | 4.89 ± 0.02 | 5.41 ± 0.03 | 5.41 ± 0.03 |
| Reg. tree | 0.40 ± 0.01 | *0.39* ± 0.01 | *0.39* ± 0.01 | **0.55** ± 0.04 | **0.56** ± 0.04 |
| GAM | 3.17 ± 0.02 | 3.36 ± 0.04 | 3.30 ± 0.03 | 3.77 ± 0.23 | 3.77 ± 0.22 |
| xgboost | 0.98 ± 0.07 | 0.92 ± 0.10 | 0.88 ± 0.09 | 1.14 ± 0.07 | 1.13 ± 0.06 |
| MLP | 3.14 ± 0.04 | 3.22 ± 0.07 | 3.20 ± 0.07 | 5.33 ± 0.14 | 5.30 ± 0.17 |
| SCIGAN | 3.05 ± 1.17 | 2.34 ± 1.84 | 1.72 ± 0.48 | 4.40 ± 4.58 | 2.08 ± 0.77 |
| DRNet | 0.97 ± 0.03 | 1.00 ± 0.04 | 1.02 ± 0.04 | 1.17 ± 0.05 | 1.16 ± 0.03 |
| VCNet | **0.29** ± 0.03 | **0.38** ± 0.02 | **0.33** ± 0.02 | *0.60* ± 0.13 | *0.60* ± 0.13 |

random.: randomized; non-unif.: non-uniformity; conf.: confounding

$N$, the true CADR $\mu(\cdot)$, and the estimated CADR $\hat{\mu}(\cdot)$ we calculate the MISE as

$$MISE = \frac{1}{N} \frac{1}{|\mathcal{T}|} \sum_{t \in \mathcal{T}} \sum_{i=1}^{N} \int_{d \in \mathcal{D}} (\mu(t, d, \mathbf{x}_i) - \hat{\mu}(t, d, \mathbf{x}_i))^2 \, \mathrm{d}d \qquad (2)$$

We use 20% of the observations in the benchmarking dataset as a holdout test set to calculate the MISE.

**Model selection.** Model selection in causal inference is challenging (Schuler et al., 2018; Curth & van der Schaar, 2023; Rolling & Yang, 2013). Several methodologies for tuning hyperparameters on observational data have been proposed, both stand-alone and accompanying an estimator. The choice of a selection procedure can have a large influence on model performance. To ensure each method performs (near-)optimally, models are selected based on the mean squared error in predicting the factual outcomes (factual selection criterion (Curth & van der Schaar, 2023)) on a validation set (10% of the units in the covariate matrix), the remaining 70% of observations are used for training models.

## 5.2 RESULTS

**Insights into the dataset.** Upon analyzing the decomposed performance of all estimators as presented in Table 1, we conclude that the TCGA-2 dataset is challenging due to dose non-uniformity, rather than dose or intervention confounding. The non-uniformity of interventions only has a small detrimental effect on model performance. Confounding of interventions has seemingly no significant effect on the performance, both for CADR estimators and supervised learning methods. The largest effect on model performance results from introducing dose non-uniformity, i.e., making some doses less likely to be observed across the population. This increases the MISE for all methods significantly when compared with scenario "$t$ confounding", in which doses are sampled from uniform distributions. The introduction of dose confounding in scenario "$d$ confounding" again does not appear to significantly impact performance. This is surprising, given the proposition of the dataset to test methods for robustness against confounding biases. Further, the non-uniformity of doses and interventions is not a causal issue, but rather related to ML challenges outside of causal inference and treatment effect modeling, such as learning from imbalanced datasets (He & Garcia, 2009; Haixiang et al., 2017). We also observe this behavior in analyzing performance for other datasets, as discussed in Section 6.

**Insights into model performance.** Comparing performance across the different estimators allows us to draw further conclusions. Estimators are only little affected by intervention non-uniformity, and not affected by their confounding. Conversely, performance might even be improved by confounding. This is counter-intuitive to the reasoning that confounding might adversely affect performance (Bica et al., 2020; Schwab et al., 2019). A possible explanation for this result is that, under confounding, units have a higher probability of being assigned exactly those doses for which CADR heterogeneity is greatest. When comparing neural architectures, CADR estimators outperform the standard MLP. However, the best-performing model is a simple regression tree. We attribute these results to the overall low heterogeneity of the dose response space in the dataset (cf. Appendix H).

**Understanding sources of performance gain.** The presented experimental results indicate that a decomposition of performance is imperative to evaluate the capabilities of CADR estimators. The observation that model performance does not degrade due to confounding, but due to non-uniform distributions of interventions

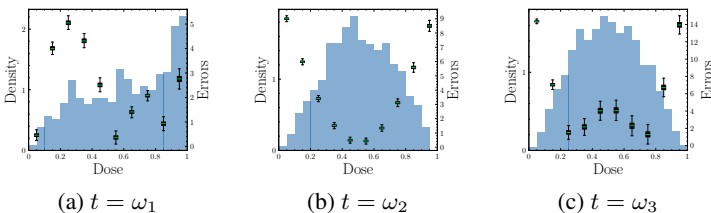

(a) $t = \omega_1$      (b) $t = \omega_2$      (c) $t = \omega_3$

Figure 4: **Distribution of errors** per intervention and dose interval the test set of the TCGA-2 dataset estimated by an MLP. A histogram of doses in the training set is added per plot in blue. Errors are correlated with dose non-uniformity, supporting that non-uniformity affects model performance.

and doses indicates that the TCGA-2 dataset is not evaluating estimators for their robustness to confounding, but rather efficiency in learning from imbalanced or limited amounts of training data (Forman & Cohen, 2004; Wang et al., 2020). This is surprising, as several ML papers use it to test estimators for robustness to confounding (Bica et al., 2020; Wang et al., 2022; Kazemi & Ester, 2024). To confirm this hypothesis, we visualize the errors in CADR estimation made by an MLP per dose interval and intervention in Figure 4, next to a histogram plot of doses in the training data. The plots show that errors in predicting CADR increase with decreasing training observations for a specific dose. This is most notable for interventions $\omega_2$ and $\omega_3$.

## 6 INSIGHTS FROM OTHER DATASETS

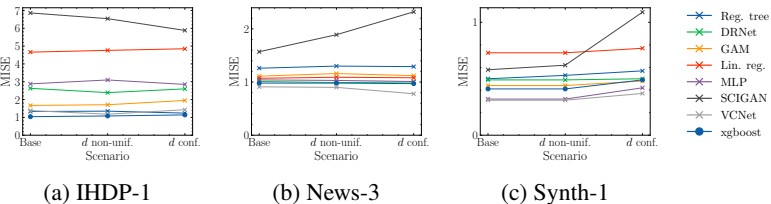

(a) IHDP-1      (b) News-3      (c) Synth-1

Figure 5: **MISE per method and dataset.** Across datasets, confounding has little adverse effects on model performance. Full results including std. errors can be found in Appendix I.

**Datasets proposed by Nie et al. (2021).** Other prominently used datasets for benchmarking CADR estimators were proposed by Nie et al. (2021) (IHDP-1, News-3, and Synth-1). We decomposed the performance of all

methods from Section 5 on all three of these datasets and present results in Figure 5). Compared to the TCGA-2 dataset, the datasets only apply a single intervention, so our decomposition does not include Scenarios 2 and 3 ($t$ non-uniformity and $t$ confounding). The datasets are confounded as both the observed doses and outcomes are conditional on the covariate matrices. Yet, as in the TCGA-2 dataset, this confounding seems to have no additional effect on model performance. Moreover, also dose non-uniformity does not affect the models. We provide a population-level distribution of doses per dataset in Appendix H, which shows that distributions are less skewed compared to TCGA-2. This explains why supervised learning techniques, especially xgboost, are performing competitively on these datasets. Therefore, experiments using these data sets only enable limited insight into a method's ability to tackle challenges inherent to CADR estimation.

**Decomposing performance under high CADR heterogeneity.** Working with synthetic datasets also allows visualizing the dose responses of individual units in the data (see Appendix H). All datasets discussed previously show little heterogeneity in the dose responses across different units. This might be a reason why supervised learning methods perform competitively against CADR estimators. We test this hypothesis by proposing a new benchmarking dataset that leverages the IHDP covariates, the "IHDP-3" dataset. IHDP-3 is distinct from the other datasets discussed in this study. Most importantly, the DGP behind the dataset assumes that there are different archetypes of units that respond distinctly to an intervention, e.g., units of one archetype might respond positively to an increased dose, while units from another archetype might respond negatively. Archetype assignment is not known ex-ante, but de-

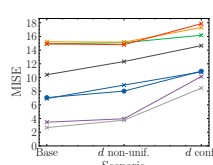

Figure 6: **MISE per method on the IHDP-3 dataset.** Strong increase in error per method attributed to confounding.

terministic to comply with strong ignorability. Confounding is introduced by sampling from a beta-distribution with a mode conditional on a unit's archetype. For the technical details of the dataset, we refer to Appendix G. The increase in heterogeneity of CADR is visualized in Appendix H.

We present the decomposed performance of CADR estimators on the IHDP-3 dataset in Figure 6 and in Appendix I. Compared to the other datasets discussed in our study, we see a significant adverse effect of dose confounding on model performance, which is evident across all estimators. Similarly, results reveal the importance of model architecture, given the estimation problem. Training individual networks per dose strata, as implemented in DRNet, leads to errors that are only comparable to linear regression, indicating that the neural architecture is not capable of handling high degrees of heterogeneity in CADR across units. Some neural architectures, especially VCNet and the standard MLP, outperform other methods in both the randomized scenario and under dose non-uniformity. The results further indicate that VCNet's varying coefficient structure aids successful confounding adjustment.

# 7  CONCLUSIONS AND FUTURE RESEARCH

This paper aims to understand the nature of conditional average dose response (CADR) estimation and to provide tools to decompose estimator performance along challenges inherent to the field. We have provided a unifying problem formulation using a single-world intervention graph (SWIG) and conceptualized synthetic data-generating processes (DGPs) typically used to evaluate estimator performance. We have proposed a novel decomposition scheme that allows us to attribute estimator performance to five challenges: The non-linearity of response surfaces, confounding of interventions and doses, and their non-uniform distributions. From our experiments, we conclude that the inherent challenges are still poorly understood. Established benchmarks are challenging predominantly due to non-uniform distributions of interventions and doses, and non-linear response surfaces. Confounding, on the contrary, seems to have little effect on model performance on these datasets. Additionally, by proposing a novel DGP and benchmarking dataset (IHDP-3), we show that confounding is a challenge for estimators only when the heterogeneity of CADR is high.

Our results show critical limitations in existing ML research on CADR estimation and suggest that more research is needed to develop benchmarks that accurately test the capabilities of estimators. We hence encourage researchers to adopt our decomposition for any future research. Additionally, we provide three further takeaways for researchers and practitioners:

**(1) Confounding can materialize in various ways.**    There is no clear-cut definition of how confounding might materialize in a DGP and previous works have not clarified how confounding is making a CADR estimation challenging. Our experiments show that the confounding in most previously established datasets does not pose a challenge. Further research is needed to understand and quantify when tailored methods are needed.

**(2) Several challenges exist in CADR estimation.**    In our experiments on established datasets, neither confounding by intervention type nor by dose have large negative impacts on model performance. Instead, the non-uniformity of doses has the largest effect. This contrasts with the claims in which these benchmarks test for robustness against (any type of) confounding and reveals that typical DGPs introduce more than just one challenge to CADR estimation. The finding is especially relevant as the non-uniformities of intervention types and doses are not causal problems. A potential future research direction might hence be the adoption of methodologies such as data-efficient ML (Olson et al., 2018; Mirzasoleiman et al., 2020).

**(3) Supervised estimators might be appropriate to model CADR.**    Finally, our results reveal that standard supervised learning methods might achieve competitive performance in estimating CADR. This supports takeaway (1) and calls for comparing any future CADR estimators against a complete set of established benchmarking methods, such as gradient-boosted trees. Improving transparency might help practitioners gain trust in ML CADR estimators and aid future adoption.

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

## A  NOTATION

We summarize the relevant notation to our paper below. Random variables are denoted in capital letters, with realizations of such in lowercase. Matrices are denoted in boldface. Realizations of a variable at a certain position, are denoted with the position as subscript.

| | |
|---|---|
| $\mathcal{X} \in \mathbb{R}^m$ | Covariate space |
| $m$ | Size of $\mathcal{X}$ (number of covariates) |
| $\mathcal{T} \in \{\omega_1, \ldots, \omega_k\}$ | Intervention space |
| $k$ | Number of possible interventions |
| $\mathcal{D} \in \mathbb{R}$ | Dose space |
| $\mathcal{Y} \in \mathbb{R}$ | Outcome space |
| $\mathbf{X} \in \mathcal{X}$ | Covariates (random variable) |
| $T$ | Interventions (random variable) |
| $D : \mathcal{T} \to \mathcal{D}$ | Potential dose function (function-valued random variable) |
| $Y : (\mathcal{T}, \mathcal{D}) \to \mathcal{Y}$ | Potential outcome function (function-valued random variable) |
| $\mu(t, d, \mathbf{x})$ | Conditional average dose response ($\mathbb{E}[Y(t, d)|\mathbf{X} = \mathbf{x}]$) |

## B  METHODOLOGY DETAILS

We illustrate the non-uniformity and confounding of doses in Figure 7, using a toy example with two individual units being assigned a dose, sampled from a probability distribution. This visualization is generalizable and serves to explain any effects on interventions as well.

In the base scenario, every dose in $\mathcal{D} = [0, 1]$ is equally likely assigned to any of the units. Under non-uniformity, some doses are more likely to be assigned, specifically lower and higher ones, whereas medium doses around 0.5 are less likely. The individual distributions of the dose per unit are equivalent to the joint distribution. In the case of confounded doses, the joint distribution across the two units is the same as under non-uniformity but individual distributions differ. Specifically, Unit 1 is assigned lower doses on average, whereas Unit 2 is assigned higher doses.

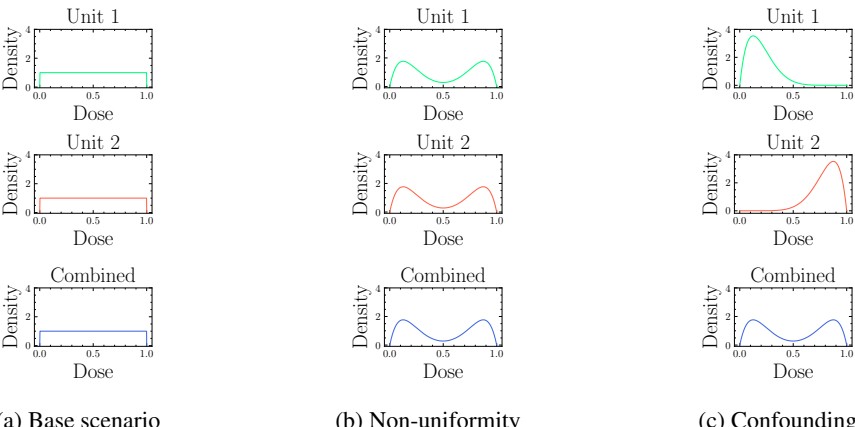

(a) Base scenario    (b) Non-uniformity    (c) Confounding

Figure 7: **Impacts of non-uniformity and confoundedness** on dose distributions per unit and per population in the case of two observed units and for doses only.

## C   STRONG IGNORABILITY IN THE DOSE RESPONSE SETTING

The strong ignorability assumption (Rosenbaum & Rubin, 1983) is typically posed for estimating treatment effects, so in the setting of binary interventions. Below we provide definitions all all components of this assumption that match the continuous case as tackled in our paper:

**Assumption 1.** *(Consistency) The observed outcome $Y_i$ for a unit $i$ that was assigned intervention $t_i$ and dose $d_i$ is the potential outcome $Y_i(t_i, d_i)$.*

**Assumption 2.** *(No hidden confounders) The assigned intervention $T$ and dose $D$ are conditionally independent of the potential outcome $Y(t, d)$ given the covariates $\mathbf{X}$, so $\{Y(t, d) | t \in \mathcal{T}, d \in \mathcal{D}\} \perp\!\!\!\perp (T, D) | \mathbf{X}$*

**Assumption 3.** *(Overlap) Every unit has a greater-than-zero probability of receiving any possible combination of intervention and dose, so $\forall t \in \mathcal{T} : \forall d \in \mathcal{D} : \forall \mathbf{x} \in \mathcal{X}$ with $\mathbb{P}(\mathbf{x}) > 0 : 0 < \mathbb{P}((t, d) | \mathbf{x}) < 1$*

## D   ML ESTIMATORS FOR CONDITIONAL AVERAGE DOSE RESPONSES

We will introduce the three CADR estimators considered in our paper in detail:

**DRNet (Schwab et al., 2019).**   DRNet uses a multitask learning approach (Caruana, 1997) to estimate conditional average dose responses. The method trains a head network per individual intervention on a set of shared layers, as motivated by Shalit et al. (2016) in the binary intervention setting. Per intervention, the method partitions the dose space $\mathcal{D}$ into a set of strata. For each strata, another individual network is trained to infer the dose response. The architecture can further be combined with regularization terms during training, to overcome potential covariate shifts between different interventions.

**SCIGAN (Bica et al., 2020).**   SCIGAN assumes that there is a shift in covariates between different levels of intervention and dose. This shift leads to non-adjusted estimators overfitting the training data. At the core of SCIGAN is a specialized generative adversarial network (GAN) structure, which attempts to generate the outcomes of counterfactual interventions and doses per unit. Creating those counterfactual observations removes the covariate shift, and allows any estimator to learn an unbiased dose response model.

**VCNet (Nie et al., 2021).**   VCNet proposes a varying-coefficient architecture (Hastie & Tibshirani, 1993). The method trains a neural network that has a network structure varying in the assigned dose per unit, reinforcing the influence of the dose on the predicted outcome. Nie et al. (2021) combine this architecture with estimating the generalized propensity score of the dose to calculate estimates of the *average* dose response through an approach proposed as "functional targeted regularization", as motivated and inspired by Shi et al. (2019). In our experiments, we do not use this part of the architecture and focus on estimating the conditional average dose response.

# E  PSEUDOCODE

We provide pseudocode for replicating our approach for an arbitrary machine-learning method and data-generating process:

---

**Algorithm 1:** Performance decomposition

---

**Require:** Covariate matrix $\mathbf{X}$, Intervention assignment function $a_t(\cdot)$, Dose assignment function $a_d(\cdot)$, Outcome function $\mu(\cdot)$, Machine learning method $\mathcal{M}$, Performance metric $P(\cdot)$

**Result:** Decomposed performance of $\mathcal{M}$ on DGP specified by $\mathbf{X}$, $a_t$, $a_d$ and $\mu$

```
# 0) initialization
```
(1) sample $ids_{train}$, $ids_{test}$
(2) $T_{rand} \leftarrow$ sample at random from $\{\omega_1, \ldots, \omega_k\}$     `# Get intervention vectors`
(3) $T_{conf} \leftarrow a_t(\mathbf{X})$
(4) $T_{non-u} \leftarrow$ shuffle $T_{conf}$
(5) $D_{rand} \leftarrow$ sample at random from $[0,1]$     `# Get dose vectors`
(6) $D_{conf} \leftarrow a_d(\mathbf{X}, T_{conf})$
(7) $D_{non-u} \leftarrow shuffle(D_{conf})$

```
# 1) eval under random interventions and doses
```
(8) $Y \leftarrow \mu(\mathbf{X}, T_{rand}, D_{rand})$     `# Get outcomes`
(9) train $\mathcal{M}$ on $(Y, \mathbf{X}, T_{rand}, D_{rand})$ using $ids_{train}$
(10) calculate $P(\mathcal{M}, Y, \mathbf{X}, T_{rand}, D_{rand})$ using $ids_{test}$     `# Calculate performance`

```
# 2) eval under intervention non-uniformity and random doses
```
(11) $Y \leftarrow \mu(\mathbf{X}, T_{rand}, D_{rand})$     `# Get outcomes`
(12) train $\mathcal{M}$ on $(Y, \mathbf{X}, T_{non-u}, D_{rand})$ using $ids_{train}$
(13) calculate $P(\mathcal{M}, Y, \mathbf{X}, T_{non-u}, D_{rand})$ using $ids_{test}$     `# Calculate performance`

```
# 3) eval under intervention confounding and random doses
```
(14) $Y \leftarrow \mu(\mathbf{X}, T_{conf}, D_{rand})$     `# Get outcomes`
(15) train $\mathcal{M}$ on $(Y, \mathbf{X}, T_{conf}, D_{rand})$ using $ids_{train}$
(16) calculate $P(\mathcal{M}, Y, \mathbf{X}, T_{conf}, D_{rand})$ using $ids_{test}$     `# Calculate performance`

```
# 4) eval under intervention confounding and dose
 non-uniformity
```
(17) $Y \leftarrow \mu(\mathbf{X}, T_{conf}, D_{non-u})$     `# Get outcomes`
(18) train $\mathcal{M}$ on $(Y, \mathbf{X}, T_{conf}, D_{non-u})$ using $ids_{train}$
(19) calculate $P(\mathcal{M}, Y, \mathbf{X}, T_{conf}, D_{non-u})$ using $ids_{test}$     `# Calculate performance`

```
# 5) evaluate under intervention confounding and dose
 confounding
```
(20) $Y \leftarrow \mu(\mathbf{X}, T_{conf}, D_{non-u})$     `# Get outcomes`
(21) train $\mathcal{M}$ on $(Y, \mathbf{X}, T_{conf}, D_{conf})$ using $ids_{train}$
(22) calculate $P(\mathcal{M}, Y, \mathbf{X}, T_{conf}, D_{conf})$ using $ids_{test}$     `# Calculate performance`

---

# F OVERVIEW OF ESTABLISHED ESTIMATORS AND BENCHMARKING DATASETS

When studying previously established dose response estimators, we find that methods have typically been evaluated on a small selection of benchmark datasets (cf. Table 2). Similarly, each paper does not compare performance against a complete set of benchmarking methods.

Table 2: **Overview of dose response estimators**

| Method | Paper | Type | Multi. treat. | Benchmark datasets | Benchmark methods | Description |
|---|---|---|---|---|---|---|
| DRNet | Schwab et al. (2019) | CA | ✓ | TCGA-1 MVICU-1 News-1 | BART (Chipman et al., 2010) Causal Forest (Wager & Athey, 2018) GANITE (Yoon et al., 2018) HIE (Hirano & Imbens, 2004) kNN (Cunningham & Delany, 2021) MLP TARNET (Shalit et al., 2016) | Multi-head neural network with separate head network per treatment. Per treatment, separate head network per dose interval. |
| SCIGAN | Bica et al. (2020) | CA | ✓ | TCGA-2 MVICU-2 News-2 | DRNet HIE MLP | GAN architecture to remove training data confounding. |
| VCNet | Nie et al. (2021) | A | ✗ | IHDP-1 News-3 Synth-1 | BART Causal forest Dragonnet (Shi et al., 2019) DRNet HIE | Varying coefficient network combined with targeted regularization. |
| ADMIT | Wang et al. (2022) | A | ✗ | TCGA-2 News-1 Synth-1 | DRNet EBCT (Tübbicke, 2021) HIE SCIGAN VCNet | Representation balancing based on weighted populations. |
| TransTEE | Zhang et al. (2022) | CA | ✓ | TCGA-2 | DRNet SCIGAN TARNet VCNet | General purpose transformer architecture for intervention response estimation. |
| CRNet[†] | Zhu et al. (2024) | CA | ✓ | IHDP-2 News-4 Synth-2 | Causal forest CBGPS (Fong et al., 2018) DRNet HIE SCIGAN VCNet | Contrastive representation balancing. |
| GIKS[†] | Nagalapatti et al. (2024) | CA | ✗ | IHDP-1 News-3 TCGA-2 | DRNet TARNet TransTEE | Adjusted learning objective based on weighted observations. |
| ACFR | Kazemi & Ester (2024) | CA | ✗ | TCGA-2 News-2 | ADMIT DRNet HIE MLP SCIGAN VCNet | Representation balancing through adversarial learning. |

[†]: No code base available, u.n.: unnamed, CA: Conditional average, A: Average, HIE: Hirano-Imbens estimator, MLP: multilayer perceptron

An overview of the proposed datasets for benchmarking dose response estimators can be found in Table 3 below, presenting the dimensionality of the covariate space, as well as the cardinality of the intervention space, and the presence of both intervention and dose confounding.

Table 3: **Benchmarking datasets for DR estimators**

| Dataset | Paper | Dim($\mathbf{x}$) | $\lvert T \rvert$ | $t$ confounding | $d$ confounding |
|---------|-------|--------|----|----|----|
| MVICU-1 | | (8040, 49) | 3 | ✓ | ✗ |
| News-1 | Schwab et al. (2019) | (5000, 2870) | {2,4,8,16} | ✓ | ✗ |
| TCGA-1 | | (9659, 20531) | 3 | ✓ | ✗ |
| MVICU-2 | | (8040, 49) | 2 | ✓ | ✓ |
| News-2 | Bica et al. (2020) | (5000, 2870) | 3 | ✓ | ✓ |
| TCGA-2[*] | | (9659, 4000) | 3 | ✓ | ✓ |
| News-3[*] | | (2993, 498) | 1 | n.a. | ✓ |
| IHDP-1[*] | Nie et al. (2021) | (747, 25) | 1 | n.a. | ✓ |
| Synth-1[*] | | (700, 6) | 1 | n.a. | ✓ |
| IHDP-2[†] | | (2993, 498) | 1 | ✓ | ✓ |
| News-4[†] | Zhu et al. (2024) | (747, 25) | {2,4,8,16} | ✓ | ✓ |
| Synth-2[†] | | (3000, 100) | {1,2,5,10} | ✓ | ✓ |
| IHDP-3[*] | *This paper* | (747, 25) | 1 | n.a. | ✓ |

[*]: Considered in this paper, [†]: No code base available, *S*: synthetic, *R*: real, *n.a.*: not applicable

# G    THE IHDP-3 DATASET

This proposal of a new dataset for benchmarking CADR estimators, the "IHDP-3" dataset, is motivated by the dose response heterogeneity (or the lack thereof) in previously established datasets. To our surprise, most estimators were not, or only a little affected by the presence of confounders in the DGP. Conversely, in those datasets, the biggest challenge in CADR estimation has been the non-uniform distribution of doses (cf. Section 5 and 6).

The IHDP-3 dataset leverages the same covariate matrix as the IHDP-1 dataset (Nie et al., 2021), notably the covariates used in the study of Hill (2011), but the assignment of intervention variables, as well as the outcome calculation differ significantly.

We consider a scenario with only one distinct intervention. Yet, different from previous datasets, the response to this intervention has higher heterogeneity in the covariates of a unit. Specifically, there are four different archetypes of responses to the intervention. We assign every unit in the covariate matrix to one of these archetypes based on their realization of binary covariates `b.marr` and `mom.lths`. The CADR per archetype is given in Table 4. Per unit, we generate the individual responses by adding normally distributed random noise.

Table 4: **CADR per archetype** of units in the IHDP-3 dataset

| AT | Dose response curve |
|----|---------------------|
| A1 | $f_1(\mathbf{x}_i, d) = 10 * (\mathbf{x}_{i,0} + 12 * d * (d - \frac{3}{4} * (\mathbf{x}_{i,1} + \mathbf{x}_{i,2}))^2)$ |
| A2 | $f_2(\mathbf{x}_i, d) = 10 * (\mathbf{x}_{i,1} + \sin(\pi * (\mathbf{x}_{i,2} + \mathbf{x}_{i,3}) * d))$ |
| A3 | $f_3(\mathbf{x}_i, d) = 10 * (\mathbf{x}_{i,2} + 12 * (\mathbf{x}_{i,3} * d - \mathbf{x}_{i,4} * d^2))$ |
| A4 | $f_4(\mathbf{x}_i, d) = \mathbf{x}_{i,0} * 3 * \sin(20 * \mathbf{x}_{i,2} * d) + 20 * \mathbf{x}_{i,3} * d - 20 * \mathbf{x}_{i,4} * d^2 + 5$ |

AT: Archetype, $\mathbf{x}_{i,j}$: Variable $j$ in the covariate vector of unit $i$

Next, we assume that doses are assigned to every observation based on their archetype, and assign every of the archetypes a modal dose in $\{\frac{1}{8}, \frac{3}{8}, \frac{5}{8}, \frac{7}{8}\}$. We then sample for every unit a factual dose from a beta distribution with the respective mode and tunable variance, following the approach first introduced by Bica et al. (2020). For a detailed DGP and the technical implementation, we refer to our source code (cf. Section J).

The resulting data has a significantly higher heterogeneity in the dose responses across units as we visualize in Appendix H.

## H    DEEP DIVE INTO BENCHMARKING DATASETS

We visualize the CADR per unit in the different benchmarking datasets in Figure 8. The heterogeneity in CADR varies along the datasets. For the previously established ones, per intervention, responses are generated from a single, but differently-parameterized function. This yields little heterogeneity in the CADR across units and might explain the good performance of supervised learning methods on these datasets. For IHDP-3, we can see higher degrees of heterogeneity with CADR depending on the archetype (cf. Appendix G). The confounding in the data significantly complicates the estimation of CADR, as seen in the detailed results (cf. Appendix I).

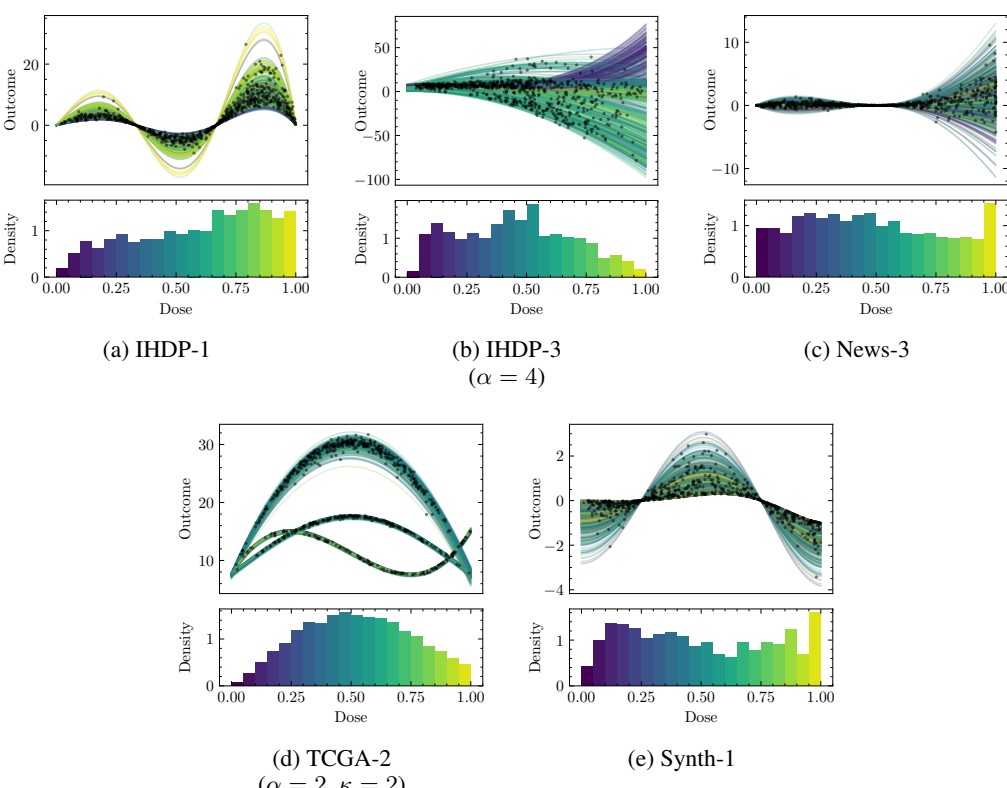

(a) IHDP-1

(b) IHDP-3
($\alpha = 4$)

(c) News-3

(d) TCGA-2
($\alpha = 2, \kappa = 2$)

(e) Synth-1

Figure 8: **Dose response space in different datasets**. Per unit in the dataset, we visualize the CADR over different interventions and doses (top plots). Accompanying, we provide a histogram of the assigned doses in the data (bottom plots). We mark the conditional response to the factual dose with a dot. The factual dose also determines the color per curve, for which the bottom plot provides a mapping. For the clarification of parameters we refer to Table 3 and the sources therein.

We further visualize confounding in the datasets by generating t-SNE plots (Hinton & Roweis, 2002; Maaten & Hinton, 2008) of their covariate spaces and color coding observations by their factual dose (cf. Figure 9). Under confounding, we expect that units different in their covariates would be assigned different factual doses. Yet, not all plots indicate this behavior. Especially in the News-3 and IHDP-1 datasets units with different factual doses cannot be separated.

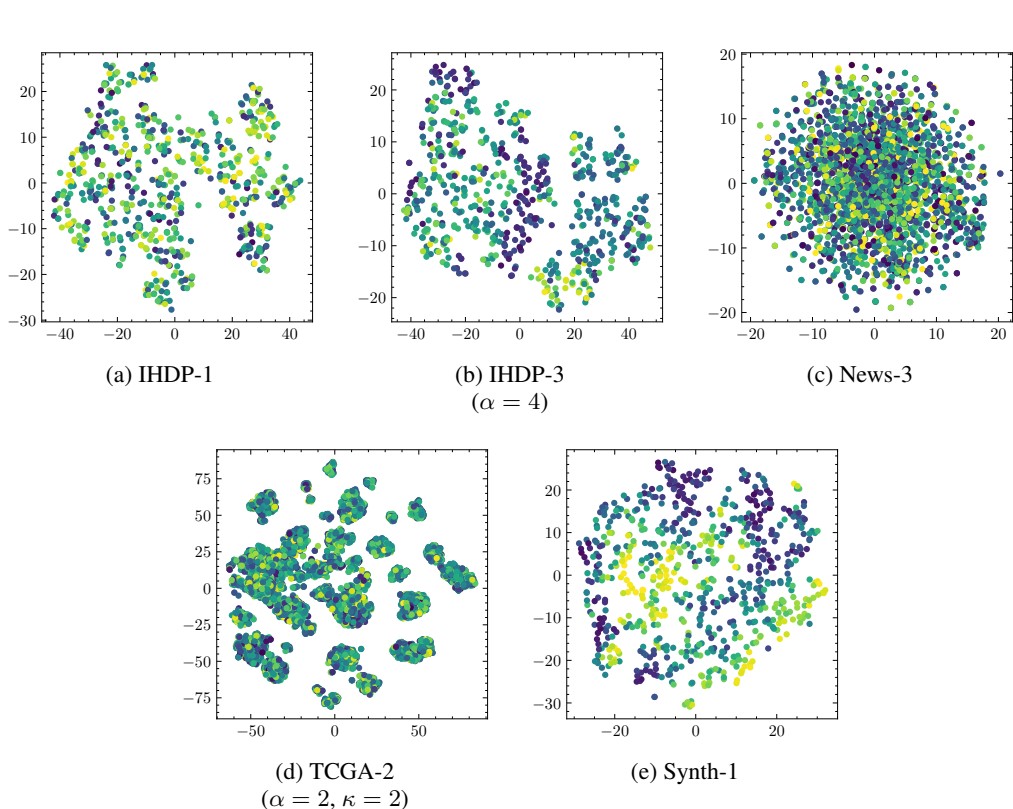

(a) IHDP-1

(b) IHDP-3
($\alpha = 4$)

(c) News-3

(d) TCGA-2
($\alpha = 2, \kappa = 2$)

(e) Synth-1

Figure 9: **t-SNE plot of covariate space per dataset**. Color corresponds to the assigned dose. We see that in the IHDP-1 and News-3 dataset observations with different assigned doses cannot clearly be separated in the t-SNE plot, indicating that the confounding in the data might not be severe. In IHDP-3, TCGA-2, and Synth-1 the separation is clearer, which might indicate a stronger effect on model performance. For the clarification of parameters we refer to Table 3 and the sources therein.

# I  RESULTS PER DATASET

Next to our detailed discussion of model performance on the TCGA-2 dataset (see Sectio 5), we provide results for every other available benchmarking dataset below:

Table 5: **Performance decomposition on IHDP-1 dataset**. According to our decomposition scheme, none of the investigated models was seriously affected by either the non-uniform distribution of doses ($d$ non-uniformity), or dose confounding ($d$ confounding). Per decomposition step (scenario), the best performing method is highlighted in **bold**, and the second best in *italics*.

| Method | *Scenario* random. | $\to d$ non-unif. $\to$ | $d$ conf. |
|---|---|---|---|
| Lin. reg. | $4.66 \pm 0.03$ | $4.76 \pm 0.06$ | $4.85 \pm 0.07$ |
| Reg. tree | $1.33 \pm 0.11$ | $1.35 \pm 0.21$ | $1.24 \pm 0.10$ |
| GAM | $1.67 \pm 0.03$ | $1.71 \pm 0.03$ | $1.95 \pm 0.13$ |
| xgboost | $\mathbf{1.04} \pm 0.10$ | $\mathbf{1.08} \pm 0.10$ | $\mathbf{1.14} \pm 0.13$ |
| MLP | $2.88 \pm 0.26$ | $3.10 \pm 0.24$ | $2.85 \pm 0.24$ |
| SCIGAN | $6.86 \pm 1.21$ | $6.54 \pm 1.07$ | $5.88 \pm 0.47$ |
| DRNet | $2.63 \pm 0.15$ | $2.39 \pm 0.06$ | $2.60 \pm 0.11$ |
| VCNet | $1.38 \pm 0.20$ | $1.18 \pm 0.20$ | $1.43 \pm 0.21$ |

random.: randomized; non-unif.: non-uniformity; conf.: confounding

Table 6: **Performance decomposition on IHDP-3 dataset**. Our decomposition scheme reveals that model performance is negatively affected by confounding. This is contrary to all other investigated datasets. Per decomposition step (scenario), the best performing method is highlighted in **bold**, and the second best in *italics*.

| Method | *Scenario* random. | $\to d$ non-unif. $\to$ | $d$ conf. |
|---|---|---|---|
| Lin. reg. | $14.91 \pm 0.21$ | $14.79 \pm 0.19$ | $17.86 \pm 0.52$ |
| Reg. tree | $6.92 \pm 0.77$ | $8.89 \pm 2.23$ | $10.82 \pm 4.19$ |
| GAM | $15.22 \pm 0.29$ | $15.16 \pm 0.32$ | $17.30 \pm 0.37$ |
| xgboost | $7.06 \pm 0.69$ | $8.01 \pm 0.57$ | $10.92 \pm 1.02$ |
| MLP | $3.47 \pm 0.23$ | $3.97 \pm 0.44$ | $10.14 \pm 0.45$ |
| SCIGAN | $10.40 \pm 2.24$ | $12.33 \pm 3.57$ | $14.65 \pm 6.24$ |
| DRNet | $14.92 \pm 0.21$ | $15.08 \pm 0.26$ | $16.17 \pm 0.16$ |
| VCNet | $\mathbf{2.68} \pm 0.35$ | $\mathbf{3.76} \pm 0.79$ | $\mathbf{8.45} \pm 0.76$ |

random.: randomized; non-unif.: non-uniformity; conf.: confounding

Table 7: **Performance decomposition on News-3 dataset**. According to our decomposition scheme, none of the investigated models was seriously affected by either the non-uniform distribution of doses ($d$ non-uniformity), or dose confounding ($d$ confounding). Per decomposition step (scenario), the best performing method is highlighted in **bold**, and the second best in *italics*.

| Method | *Scenario* | | |
|---|---|---|---|
| | random. $\rightarrow$ | $d$ non-unif. $\rightarrow$ | $d$ conf. |
| Lin. reg. | 1.07 ± 0.10 | 1.09 ± 0.11 | 1.08 ± 0.10 |
| Reg. tree | 1.26 ± 0.13 | 1.30 ± 0.10 | 1.29 ± 0.18 |
| GAM | 1.11 ± 0.08 | 1.16 ± 0.08 | 1.12 ± 0.05 |
| xgboost | *0.98* ± 0.06 | *0.98* ± 0.04 | *0.97* ± 0.05 |
| MLP | 1.04 ± 0.08 | 1.03 ± 0.12 | 1.01 ± 0.11 |
| SCIGAN | 1.57 ± 0.15 | 1.89 ± 0.22 | 2.32 ± 1.71 |
| DRNet | 1.01 ± 0.10 | 0.99 ± 0.09 | 1.00 ± 0.08 |
| VCNet | **0.91** ± 0.05 | **0.90** ± 0.04 | **0.78** ± 0.06 |

random.: randomized; non-unif.: non-uniformity; conf.: confounding

Table 8: **Performance decomposition on Synth-1 dataset**. According to our decomposition scheme, none of the investigated models was seriously affected by either the non-uniform distribution of doses ($d$ non-uniformity), or dose confounding ($d$ confounding). Per decomposition step (scenario), the best performing method is highlighted in **bold**, and the second best in *italics*.

| Method | *Scenario* | | |
|---|---|---|---|
| | random. $\rightarrow$ | $d$ non-unif. $\rightarrow$ | $d$ conf. |
| Lin. reg. | 0.73 ± 0.03 | 0.73 ± 0.03 | 0.77 ± 0.03 |
| Reg. tree | 0.50 ± 0.05 | 0.53 ± 0.12 | 0.57 ± 0.11 |
| GAM | 0.44 ± 0.03 | 0.44 ± 0.03 | 0.48 ± 0.04 |
| xgboost | 0.41 ± 0.03 | 0.41 ± 0.02 | 0.49 ± 0.04 |
| MLP | *0.32* ± 0.02 | *0.32* ± 0.03 | *0.42* ± 0.05 |
| SCIGAN | 0.58 ± 0.11 | 0.62 ± 0.09 | 1.09 ± 0.13 |
| DRNet | 0.49 ± 0.03 | 0.49 ± 0.03 | 0.50 ± 0.03 |
| VCNet | **0.31** ± 0.03 | **0.31** ± 0.03 | **0.37** ± 0.04 |

random.: randomized; non-unif.: non-uniformity; conf.: confounding

## J  IMPLEMENTATION AND HYPERPARAMETER OPTIMIZATION

All experiments were written in Python 3.9 (Van Rossum et al., 1995) and run on an Apple M2 Pro SoC with 10 CPU cores, 16 GPU cores, and 16 GB of shared memory. The system needs approximately two days for the iterative execution of all experiments. The code to reproduce all experiments and figures in our paper can be found online via https://anonymous.4open.science/r/CADR-performance-deco-8148 including a reference to the necessary covariate matrices.

For SCIGAN and VCNet, we use the original implementations provided by Bica et al. (2020) (https://github.com/ioanabica/SCIGAN) and Nie et al. (2021) (https://github.com/lushleaf/varying-coefficient-net-with-functional-tr). All remaining neural network architectures were implemented in PyTorch (Paszke et al., 2017) using Lightning (Falcon et al., 2020). Xgboost is implemented using the xgboost library (Chen & Guestrin, 2016). GAMs were implemented using the PyGAM library (Servén et al., 2018). All other methods were implemented using the Scikit-Learn library (Pedregosa et al., 2011) and the statsmodels library (Seabold & Perktold, 2010).

For TCGA-based datasets, linear regression models and GAMs were trained using the first 50 principal components of the covariate matrix to reduce computational complexity.

**Hyperparameter optimization.**  For all methods, we used a validation set for hyperparameter optimization and chose the best model in terms of validation set mean squared errors (MSE). We do so to ensure fair model comparison and isolate model performance from parameter selection procedures, as presented accompanying some estimators (Schwab et al., 2019; Bica et al., 2020). We ran a random search over the hyperparameter ranges as listed per the model below. If not specified differently, the remaining hyperparameters are set to match the specifications of the original authors. Results are not to be compared to the original papers, as the optimization scheme and parameter search ranges differ from the original records.

Table 9: Hyperparameter search range for Linear Regression:

| Parameter | Values |
|-----------|--------|
| Penalty | $\{Elasticnet, None\}$ |

Table 10: Hyperparameter search range for Regression Tree:

| Parameter | Values |
|-----------|--------|
| Max depth | $\{5, 15, None\}$ |
| Min sample split | $\{2, 5, 20\}$ |
| Min samples per leaf | $\{1, 5, 10\}$ |
| Max features per split | $\{None, \sqrt{p(\mathbf{x})}\}$ |
| Splitting criterion | $\{Gini\}$ |

Table 11: Hyperparameter search range for GAM:

| Parameter | Values |
|-----------|--------|
| Interaction type | $\{Univariate\}$ |
| Numb configurations | $\{20\}$ |

Table 12: Hyperparameter search range for xgboost:

| Parameter | Values |
| --- | --- |
| Learning rate | $\{0.01, 0.1, 0.2\}$ |
| Max depth | $\{3, 5, 7, 9\}$ |
| Subsample | $\{0.5, 0.7, 1.0\}$ |
| Min child weight | $\{1, 3, 5\}$ |
| Gamma | $\{0.0, 0.1, 0.2\}$ |
| Columns sampled per tree | $\{0.3, 0.5, 0.7\}$ |

Table 13: Hyperparameter search range for MLP:

| Parameter | Values |
| --- | --- |
| Learning rate | $\{0.0001, 0.001\}$ |
| L2 regularization | $\{0.0, 0.1\}$ |
| Batch size | $\{64, 128\}$ |
| Hidden size | $\{32, 48\}$ |
| Num steps | $\{5000\}$ |
| Num layers | $\{2\}$ |
| Optimizer | $\{Adam\}$ |

Table 14: Hyperparameter search range for SCIGAN:

| Parameter | Values |
| --- | --- |
| Hidden size | $\{32, 64, 128\}$ |
| Batch size | $\{128, 256\}$ |
| Num head layers | $\{2\}$ |
| Num dose samples | $\{5\}$ |
| $\lambda$ | $\{1\}$ |
| Optimizer | $\{Adam\}$ |

Table 15: Hyperparameter search range for DRNet:

| Parameter | Values |
| --- | --- |
| Learning rate | $\{0.0001, 0.001\}$ |
| L2 regularization | $\{0.0, 0.1\}$ |
| Batch size | $\{64, 128\}$ |
| Hidden size | $\{32, 48\}$ |
| Num dose strata | $\{10\}$ |
| Num steps | $\{5000\}$ |
| Num layers | $\{2\}$ |
| Optimizer | $\{Adam\}$ |

Table 16: Hyperparameter search range for VCNet:

| Parameter | Values |
| --- | --- |
| Learning rate | $\{0.001, 0.01\}$ |
| Batch size | $\{128, 256\}$ |
| Hidden size | $\{32\}$ |
| Num steps | $\{5000\}$ |
| Optimizer | $\{Adam\}$ |

