# OpenReview forum: "Sources of Gain: Decomposing Performance in Conditional Average Dose Response Estimation"
_ICLR.cc/2025/Conference — ICLR 2025 Conference Withdrawn Submission_

### Official Review · Reviewer_xo8C · 2024-10-15

**Soundness:** 3
**Presentation:** 3
**Contribution:** 1
**Rating:** 3
**Confidence:** 4

**Summary:**

The paper proposes a new framework for constructing (semi-) synthetic benchmark datasets for causal inference, specifically the estimation of dose-response curves with continuous treatments. In their framework, the authors distinguish between distribution shifts and confounding on both the treatment type and the dosage given. Experiments are performed using a set of established estimators and datasets from the literature.

**Strengths:**

- The paper addresses an important and relevant topic: benchmarking practices in causal inference are not as streamlined as in other ML disciplines and it is not clear how to construct good benchmarks for causal inference methods. Steps towards a streamlined benchmarking process are necessary for the field to move towards a transparent evaluation of proposed methods.
- The paper is well-written and easy to follow.

**Weaknesses:**

- Novelty: While I appreciate the general topic of the paper, I think that the novel ideas are relatively limited. As far as I understand, the proposed framework essentially disentangles the different confounding mechanisms in treatment and dosage.
- It is unclear why the proposed framework (i.e., steps 1-5 on page 5) is the best way to approach benchmarking for dose responses. It untangles the different sources of confounding but does not address other challenges for dose-response estimation. For example, lack of overlap is probably equally important when considering continuous treatments in causal inference. While there are certain connections between confounding strength and lack of overlap (e.g., a strong confounder will make overlap less likely), the framework does not address this in a principled manner. Another aspect that is missing is the complexity of the response function and treatment assignment, which can play important roles in estimator performance (see e.g., Kennedy 2023, Towards optimal doubly robust estimation of heterogeneous causal effects).
- The list of dose-response estimators considered (e.g., Table 2) is incomplete. For example, the causal forest can be used to obtain dose-response estimates. Additionally, all estimators considered are so-called "plugin" learners. In contrast, meta-learners are built upon semiparametric efficiency theory and Neyman orthogonal losses and can be combined with most regression-based approaches considered in this paper. For example, the R-learner ("Double ML") is implemented here for dose responses: https://econml.azurewebsites.net/spec/estimation/dml.html.
- I think results for the full framework on an additional dataset would strengthen the paper, given that the full framework does not apply to the datasets in Sec. 6.

**Questions:**

- The framework consists of 5 discrete steps which seem arbitrary. E.g., in the "non-uniformity" parts, the authors replace the treatment distribution but how is it generally specified? I expect the results to be highly dependent on that (e.g., setting the density to a highly concentrated distribution will result in strong overlap violations). The same holds for confounding: how should we decide on the "strength" of confounding, which is not a discrete quantity?

- How exactly are the datasets changed to fit into the framework? I assume the authors construct semi-synthetic datasets by using real covariates and creating synthetic treatments and outcomes?

---

> ### Author Response · Authors · 2024-11-24
>
> Thank you for your review and suggestions to improve our manuscript.
>
> Our proposed decomposition scheme has by no means the ambition to be the only way to decompose model performance on a dataset. However, we believe that it is an intuitive approach, motivated by real-life data-generating processes.
>
> Regarding your comment on the complexity of the dose response, we kindly refer to our response to Reviewer 5efm.

---

### Official Review · Reviewer_2PPQ · 2024-10-17

**Soundness:** 2
**Presentation:** 2
**Contribution:** 1
**Rating:** 3
**Confidence:** 4

**Summary:**

The paper performs an empirical study to better understand the performance in dose-response estimation (i.e., estimating causal effects from continuous treatments). The starting point for the paper is that benchmarking datasets for dose-response estimation missing. As such, the paper rightfully says that (1) continuous treatment is an important setting and that (2) benchmarking in causal effect estimation is overlooked (and should receive more attention). Similar benchmarking studies have been conducted by several papers (see below), however, for binary treatments and **not** continuous treatments.

Curth, A., & Van Der Schaar, M. (2023, July). In search of insights, not magic bullets: Towards demystification of the model selection dilemma in heterogeneous treatment effect estimation. In International Conference on Machine Learning (pp. 6623-6642). PMLR.

Mahajan, D., Mitliagkas, I., Neal, B., & Syrgkanis, V. (2022). Empirical Analysis of Model Selection for Heterogeneous Causal Effect Estimation. arXiv preprint arXiv:2211.01939.

Bach, P., Schacht, O., Chernozhukov, V., Klaassen, S., & Spindler, M. (2024, March). Hyperparameter Tuning for Causal Inference with Double Machine Learning: A Simulation Study. In Causal Learning and Reasoning (pp. 1065-1117). PMLR.

**Strengths:**

* Benchmarking is relevant for causal ML and especially dose-response estimation (=continuous treatments) receive to little attention.
* Clear writing
* Code is available (but I did not run the code)

**Weaknesses:**

The depth of the implications is limited. Let me give you a few examples and also make suggestions for how to improve the contribution of the work. For example, the paper correctly points out that standardized benchmarks for dose-response estimation are missing. (For comparison, there is ACIC2016+2018 for binary treatment). The paper could make (for example) a much larger contribution if it would contribute new, standardized datasets.

Likewise, the paper reports some results, but is fairly silent about meaningful implications for the field to evolve. I would love to see a paper that offers more 'implications' and 'learnings' that help researchers. A nice example is the paper from Alicia Curth, where very detailed implications are given. This might help to make the paper stronger and more meaningful (e.g., under which conditions should which estimator be used?).

The findings would be stronger if more than one dataset were used; it is likely that other datasets may lead to different results.

(Disclaimer: I used DR estimation myself but I would not have any real takeaway messages that I take from the paper. I think bechmarking studies are relevant and strongly recommend that the authors submit their work eg. to CLeaR)

Minor: Fig 2+3 imho have little value for the main paper.

**Questions:**

Q1: Why is the intervention space discrete if the interventions are supposed to be continuous? Why do you split t vs. d? (that's at least not the standard setting of DR; I like the flexibility but it may need a few additonal comments)

Q2: Table 1 reports +/-. What is this SD/SE/range? Over how many runs? (I may have overlooked this detail)

---

> ### Author Response · Authors · 2024-11-24
>
> Thank you for your review and for highlighting the relatedness with the work from Curth & Van Der Schaar (2023).
>
> You are correct that our paper does not offer immediate implications to researchers. This is mostly because we are focusing on datasets, not methods. Instead of proposing suited estimators, we rather want to draw attention to challenges in judging the performance of an estimator by benchmarking it on a single or only a few synthetic datasets.
>
> With respect to the number of datasets: We are in fact assessing the four most widely used datasets, not only TCGA-2, and additionally propose a novel dataset.
>
> Concerning Q1: As we point out, we postulate that a unit can be exposed to one of several different interventions, hence the discreteness, each with a dose that is continuous.
>
> Concerning Q2: Thanks for the remark. We will clarify this in an updated version.

---

### Official Review · Reviewer_MoZM · 2024-10-30

**Soundness:** 2
**Presentation:** 3
**Contribution:** 1
**Rating:** 3
**Confidence:** 4

**Summary:**

This paper considers the practical implications of estimating dose responses with multiple intervention categories. The main argument is that confounding is different from non-uniform treatment distributions, and the latter seems to be the main source of difficulty in commonly used benchmark datasets. If so, this is important because the community generally considers confounding to be the main challenge.

**Strengths:**

The weaknesses in the literature on estimating dose responses are clearly presented, and so is the distinction between confounding and treatment non-uniformity. Taking a closer look at the commonly used semi-synthetic benchmarks and their statistical characteristics is important especially because the community sometimes glosses over these details.

**Weaknesses:**

While the goal of decomposing the statistical challenges of estimating dose responses is highly significant, I am uncertain if this contribution is substantive. My primary concern has to do with the narrow scope of the empirical evaluations. The authors focused on the TCGA-2 semi-synthetic benchmark proposed by Bica et al. (2020). While this is definitely one of the more commonly used datasets in the literature, there is little consensus on benchmarks in the field as a whole and many papers propose their own variants.

More broadly, the claim that confounding does not matter as much as non-uniformity of treatments is not strongly supported. This is evident in Section 6, where the results for IHDP-1, News-3, and Synth-1 are much more mixed. As for TCGA-2, I was expecting an analysis of different strengths of confounding to see the level at which confounding is or is not important, compared to the other "knobs". Presently, the paper considers just one specific version of confounding. Given the generality of what we can consider "confounding", these findings are more a critique of TCGA-2 than any generalizable insight. It would have been interesting if, at the very least, one of the hyperparameters for TCGA-2 were varied to allow exploration of different confounding landscapes.

Minor point: please explain in Table 1's caption that you are listing errors (MISE).

**Questions:**

Can these findings generalize more broadly? Do you have suggestions for future benchmarks to include forms of confounding that are more challenging?

---

> ### Author Response · Authors · 2024-11-24
>
> Thank you for the assessment of our work, your feedback, critique, and suggestions for improvement.
>
> We are happy to hear that we made the importance of our work clear enough and that you agree to its relevance.
>
> You rightly mentioned in your review that some of the benchmarking datasets for CADR estimation allow altering the “strength of confounding”, especially the TCGA-2 dataset.
>
> However, it is this very aspect that has motivated our work. When examining the data-generating process behind TCGA-2, we see that increasing the strength will also increase dose non-uniformity. In our experiments, this increase has yet not changed the results from our decomposition and all of the measured performance differences were due to non-uniformity and not confounding.
>
> You are right, however, that this information is missing in the current version of the manuscript. We will add this in an updated version.

---

### Official Review · Reviewer_5efm · 2024-11-02

**Soundness:** 2
**Presentation:** 3
**Contribution:** 1
**Rating:** 3
**Confidence:** 4

**Summary:**

The paper proposes a novel decomposition scheme for benchmarking CADR estimators. For that, the authors propose alternating different steps in the treatment and dosage assignment to investigate the effect of confounding on model performance. Further they provide experiments on widely-used benchmark datasets giving novel insights for existing CADR methods.

**Strengths:**

-	The paper tackles an important problem in causal inference by trying to propose a unifying approach for model evaluation (here for CADR estimation). In principle, I think this is a widely understudied issue as most papers just design their own semi-synthetic datasets, making it harder to evaluate the fairness of model comparisons.
-	The paper is nicely written and easy to follow

**Weaknesses:**

-	The contribution of the paper is only very limited. While I think it is nice to have a paper summarizing challenges in CATE/CADR evaluation and propose best practices, I think the added value is not sufficient for an ICLR paper. This is mainly a benchmarking paper and the setup of the scheme seems pretty arbitrary.
-	The interpretation of key parts of their framework could be clearly improved.
    -  The main goal is to “decompose the source of gain” in CADR estimation. However, the main focus of their DGP decomposition is only around the treatment and dosage assignment. However, if there is no heterogeneity in the dose-response curve, stronger dependence between assignment and X does not necessarily increase confounding since Y does not (strongly) depend on X. Thus, non-uniformity can be obviously even more challenging than the “confounded” setting because of increased data sparsity in some areas with low values of the marginal densities of the doses. While the authors acknowledge that and propose a dataset with increased heterogeneity in the DR curve, this finding should imply that for disentangling the performance drivers of CADR estimators such framework needs to jointly vary treatment/dosage assignment and heterogeneity of the DR curve and probably focus less on a distinction between non-uniformity and confounding.
    -	The evaluated benchmarks are also more dependent on the true DR curve than on the treatment/dosage assignment (e.g., DRNet and VCNet strongly depend on the “nodes” / “strata”, DRNet only does balancing on the treatment level so it’s not designed for dosage bias, VCNets functional targeted regularization is designed for optimizing ADR estimation instead of CADR). This could be more discussed and, thus, investigating the complexity and heterogeneity of the DR curve would probably give more insights into the advantages of such methods.

**Questions:**

-	Why is the focus of the scheme on the treatment/dosage assignment and not the DR function?

---

> ### Author Response · Authors · 2024-11-24
>
> Thank you for reviewing our submission.
>
> We appreciate your positive feedback concerning the soundness and presentation of our manuscript.
>
> While we agree with the assessment that the heterogeneity of the dose response has a major role in determining the complexity and hardness of CADR estimation, we would argue that the ground truth dose response is typically specific to an application.
>
> Our decomposition scheme is not meant to identify a single best-performing estimator, as such an estimator likely does not exist.
>
> This is also why we did not decide to decompose the dose response any further.

---

### Official Review · Reviewer_ys6X · 2024-11-02

**Soundness:** 4
**Presentation:** 4
**Contribution:** 4
**Rating:** 10
**Confidence:** 4

**Summary:**

Estimation of Conditional Average Dose Responses (CADR) is a challenging problem. The performance of a CADR estimator depends on its ability to cope with non-uniformity of intervention and/or dose assignment, and the presence of confounding variables that impact the outcome as well as the intervention and/or dose assignment. Which of these challenges is more or less important in a particular empirical setting depends on the nature of the underlying Data Generating Process (DGP). In ML, the performance of CADR estimators are typically assessed by benchmarking performance on synthetic or semi-synthetic datasets. These facts motivate a decomposition of the performance of CADR estimators on synthetic/semi-synthetic data by sequentially removing different challenges by altering the underlying DGP. In particular, one first assesses performance under the unaltered DGP with confounding of both dose and intervention. Then under confounded interventions but with unconfounded and non-uniform doses, then confounded and unconfounded uniform doses, unconfounded non-uniform interventions and unconfounded uniform doses, and finally uniform and unconfounded doses and interventions.

By assessing performance under this sequence of altered DGPs, one can evaluate the impact of each of these distinct challenges (confoundedness and non-uniformity of intervention and dose) on estimator performance. If performance of an estimator improves substantially after one challenge is removed, then one can conclude that this challenge is responsible for a substantial degradation of performance for that estimator in that dataset. Thereby, one can assess a) whether some estimators are better able to deal with a particular one of the challenges, and b) how much difficulty each of these challenges poses in general in a particular dataset.

The authors then apply their methods to commonly used benchmarking datasets. Among other things, they demonstrate that in the TCGA-2 dataset confounding does not pose a significant challenge for the methodsthey apply, whereas non-uniformity of the dose has a much greater negative impact on performance.

The authors also find that modern neural-network methods do not exhibit superior performance in standard datasets, and they suggests that this may be because the degree of heterogeneity in the CADR is insufficiently high for the flexibility of these methods to be a substantial advantage. In resoponse, they develop a new synthetic dataset with greater heterogeneity.

**Strengths:**

I really like this paper. I think it is insightful, thorough, well-explained, and I find the authors' arguments convincing. I believe the proposed benchmarking approach is viable and I hope it catches on.

**Weaknesses:**

Another feature of a DGP that can pose a substantial challenge for CADR estimation is weak covariate overlap. By `weak overlap' I mean cases in which $0<P((t,d)|x)<1$ but $P((t,d)|x)$ is close to zero or one for some values of $x$, $t$, and $d$. As with non-uniformity of dose and intervention, this problem can be hard to disentangle from the problem of confounding. I wondered whether the individual impact of weak overlap could be assessed by further subdividing the five scenarios on page 5.

**Questions:**

As I mention above, I wondered whether the methods presented here could be extended to analyze the particular impact of weak overlap separately from the problem of confounding. Do you think that would be possible?

---

> ### Author Response · Authors · 2024-11-24
>
> Thank you for the thorough review, feedback, and positive assessment of our work.
>
> We agree with your critique regarding weak covariate overlap.
> Indeed, this topic is closely related to the non-uniformity of interventions and doses.
>
> However, incorporating an assessment of the impact of weak overlap into our decomposition scheme is difficult, as most covariate matrixes are sampled from real datasets and hence limited in size. Further subsetting of these datasets might make limited sense for such an assessment.
>
> Nevertheless, an intriguing future research direction would be to assess the sample efficiency of CADR estimators.

---

### Author Response · Authors · 2024-11-24

We want to thank the editors and reviewers of ICLR 2025 for their reviews and feedback on our manuscript. Those have been valuable pointers on how to improve and update our work to enhance clarity, contribution, and impact.

We as authors have decided that, based on the reviews, ICLR is not an optimal format for the content and contributions of our work and are thus announcing that we will withdraw our paper.

---

### Note · Authors · 2024-11-24

I have read and agree with the venue's withdrawal policy on behalf of myself and my co-authors.